# Animal Research, Accountability, Openness and Public Engagement: Report from an International Expert Forum

**DOI:** 10.3390/ani9090622

**Published:** 2019-08-29

**Authors:** Elisabeth H. Ormandy, Daniel M. Weary, Katarina Cvek, Mark Fisher, Kathrin Herrmann, Pru Hobson-West, Michael McDonald, William Milsom, Margaret Rose, Andrew Rowan, Joanne Zurlo, Marina A.G. von Keyserlingk

**Affiliations:** 1Animal Welfare Program, Faculty of Land and Food Systems, University of British Columbia, Vancouver, BC V6T 1Z4, Canada; 2Department of Clinical Sciences, Swedish University of Agricultural Sciences, 75007 Uppsala, Sweden; 3Principal Adviser, Animal Welfare, Ministry for Primary Industries, Wellington 6140, New Zealand; 4The Center for Alternatives to Animal Testing (CAAT), Johns Hopkins Bloomberg School of Public Health, Baltimore, MD 21205, USA; 5School of Sociology and Social Policy, University of Nottingham, Nottingham NG7 2RD, UK; 6Maurice Young Centre for Applied Ethics, University of British Columbia, Vancouver, BC V6T 1Z4, Canada; 7Department of Zoology, University of British Columbia, Vancouver, BC V6T 1Z4, Canada; 8Director Research Governance, South Eastern Sydney and Illawarra Shoalhaven Local Health Districts, Conjoint Professor, University of New South Wales, Sydney 2036, Australia; 9The Humane Society of the United States, Washington, DC 20037, USA

**Keywords:** animal ethics, animal experimentation, animal welfare, governance, policy, public engagement

## Abstract

**Simple Summary:**

The issues of openness, transparency and public engagement about animal research have taken focus in several different countries in recent years. This paper gives an account of a two-day-long expert forum that brought together policy experts and academics from Australia, Canada, Germany, New Zealand, Sweden, the United Kingdom and the United States. The aim was to share current governance practices regarding openness and transparency of animal research and to brainstorm ideas for better public engagement. The facilitated conversations were transcribed and analysed to create this report and recommendations that encourage international policy-makers and other stakeholders to engage in genuine dialogue about the use of animals in research.

**Abstract:**

In November 2013, a group of international experts in animal research policy (n = 11) gathered in Vancouver, Canada, to discuss openness and accountability in animal research. The primary objective was to bring together participants from various jurisdictions (United States, Sweden, Australia, New Zealand, Germany, Canada and the United Kingdom) to share practices regarding the governance of animals used in research, testing and education, with emphasis on the governance process followed, the methods of community engagement, and the balance of openness versus confidentiality. During the forum, participants came to a broad consensus on the need for: (a) evidence-based metrics to allow a “virtuous feedback” system for evaluation and quality assurance of animal research, (b) the need for increased public access to information, together with opportunities for stakeholder dialogue about animal research, (c) a greater diversity of views to be represented on decision-making committees to allow for greater balance and (d) a standardized and robust ethical decision-making process that incorporates some sort of societal input. These recommendations encourage aspirations beyond merely imparting information and towards a genuine dialogue that represents a shared agenda surrounding laboratory animal use.

## 1. Introduction

The use of animals in research presents an interesting balance between the need for openness and confidentiality. Previous work has shown that during a crisis transparent communications lead to higher levels of public trust [1]. Sharing details about animal-based research practices can add legitimacy to such research by increasing public trust. Historically, sharing details about animal research was thought to put institutions at increased risk of exposure to animal rights activism [2,3]. However, such targeting by animal rights activists is currently at a worldwide low [4]. There is also evidence that public acceptance of animal research has continually declined over the past two decades [5,6], perhaps associated with a decrease in public trust of animal research.

In 2012, the University of British Columbia (UBC) became the first University in North America to publish the species and numbers of animals used in research on campus in a given year, along with information on the level of invasiveness of the procedures used on the animals listed. While UBC made the move to increased openness, it has received criticism because of the controlled nature of the information sharing, and some argued that this effort does not go far enough in terms of promoting public engagement about animal research [7,8]. Practices of “selective openness” seem to be the norm within the culture of the animal research community [3].

Calls for public governance of science have become increasingly prevalent for at least the last 15 years [9,10,11,12]. Community engagement in the governance of controversial research is now widely acknowledged to be an important part of the public governance of science [13,14,15]. Efforts toward public engagement generally seek to democratize science policy-making by expanding expert-driven conversations to include lay citizens [16]. Despite the public attention that this issue has received, and the recognition of a need for public accountability in science generally [17], little consensus has developed on how to better engage members of the public on issues related to animal-based research.

Meaningful public engagement would seem to be unrealistic without information sharing. Calls for greater openness have long been central to the discourses of critics: what has now changed is the extent to which this agenda is being embraced by the scientific community, at least in the UK [18]. For example, the Basel Declaration [19] (signed on 29 November 2010 by a group of 80 biomedical scientists from Sweden, Germany, Switzerland, France and Great Britain following a meeting to discuss issues related to animal research) committed to “promote the dialogue concerning animal welfare in research by transparent and fact-based communications to the public.” Similarly, at the 8th World Congress on Alternatives and Animal Use in the Life Sciences, held in Montreal in 2011, attendees passed the Montreal Declaration calling for an “increase in the transparency of the translation of animal-based research” [20]. In 2012, over 40 organizations involved in bioscience signed The UK Concordat on Openness on Animal Research and committed to be more open about the ways in which animals are used in scientific, medical and veterinary research. Signatories now number 120. As part of the process, members of the UK public were consulted on the four commitments in the Concordat: (1) “we will be clear about how, when and why we use animals in research”, (2) “we will enhance our communication with the media and the public about our research using animals”, (3) “we will be proactive in providing opportunities for the public to find out about research using animals” and (4) “we will report our progress annually and share our experiences” [21]. Similar initiatives have been signed in Belgium, Spain and Portugal [22,23,24]. There was also public dialogue on openness and animal research, which explored the public’s views on openness and transparency in animal research and found that “The public want the sector to demonstrate its commitment to openness by creating greater scrutiny of itself” [25]. In response to these initiatives, some authors have called for more discussion of precisely what openness can be expected to achieve [18].

Given the difficulties that arise when trying to balance openness and confidentiality, the local nature of this move toward increased openness at UBC provided an incentive for UBC researchers to host a meeting with international experts. The aim of this meeting was to share best practices and discuss international differences and challenges, and to develop a vision for increasing openness. The meeting also provided an opportunity to articulate why a move towards increased openness is important in terms of building an ethical culture of research and to explore options for more meaningful dialogue about animal research with the broader public.

## 2. Materials and Methods

### 2.1. Recruitment of Expert Participants

Participants were recruited via email on the basis of their academic expertise or their current role in policy-making. All participants were required to have working knowledge of the governance of animal research in their country of residence, and before the forum each participant provided a written summary of their governance system. Of the 11 participants, 7 were primary involved in academia, and 4 in policy, but all had previously contributed to either regional or national policy-making processes.

The aim was to recruit participants who represent those countries with a range of governance systems for the use of animals in science, so participants from the USA, UK, Canada, Sweden, Australia, New Zealand and Germany were recruited. Together, these seven countries represent a variety of governance styles ranging from fully legislated (e.g., UK), to partially legislated (e.g., USA) to non-legislated (e.g., Canada), a range of national systems, from state governance (e.g., Australia, Germany) to national governance (e.g., UK), as well as varying degrees of openness, from more open (e.g., Sweden) to less open (e.g., USA). More participants could have been recruited, from within these countries and from other nations. However, the organizers wanted to balance the diversity of viewpoints and ensure the ability for constructive discussion over a two-day forum.

### 2.2. Discussion Groups and Facilitation

Forum discussions occurred over a two-day period and were divided into small group sessions and plenary sessions. The topics to be discussed were selected by the organizers based on the pre-meeting summaries submitted by the participants. Each topic was presented to the whole group at the beginning of each session. The group then broke out into two smaller groups for discussion. Facilitators moderated the discussion in the small groups, and scribes were present to record the conversation. The scribes wrote notes on a flip chart so that all group members see if the conversation was faithfully captured. Small-group discussions lasted between 45 and 60 min. Once finished, the small groups reconvened in plenary, and the scribes and facilitators gave feedback to the whole group about what was discussed. This feedback formed the basis of a plenary discussion lasting approximately 60–90 min. All plenary sessions were audio-recorded and transcribed for analysis.

### 2.3. Analysis

Inductive thematic content analysis of the plenary sessions was carried out by the first author. The small-group sessions had scribes present to take notes of the topics discussed—these notes were typed up and referred to as part of the analysis. The small groups also gave feedback about their discussions during the plenary sessions to reduce the risk the plenary sessions missed any key points.

Thematic content analysis followed the process described by Coffey and Atkinson [26] and Burnard et al. [27]. In brief, the transcripts were read and, line by line, labels were assigned to sections (i.e., sentences or phrases) of the text. This method of qualitative coding has been described as the use of “tags or labels for assigning units of meaning to the descriptive or inferential information compiled during a study” [28]. The process of tagging was iterative, and some tags were changed as more tags were added. The tags were then grouped together into themes to describe the major concepts raised in the discussions. Again, this was an iterative process, and on occasion tags or themes were altered to improve fit and more faithfully represent the content of the conversations. The structure of the forum produced an environment for synergistic discussion among the participants. While direct quotes are used in this paper, they will not be attributed to a specific individual because the dialogue was co-constructed, meaning that several participants often contributed to the formation of each point. Forum discussion focused on ethics review committees, institutional culture and policy and oversight.

## 3. Results and Discussion

### 3.1. Ethics Review Committee

Participants discussed the structure and functioning of the protocol review committees responsible for approving animal-based research protocols in each of the countries. There was general desire for increased openness and public access to animal research protocols, as well as a desire for balanced selection of committee members and recruitment methods that reduce biases and consider other perspectives.

#### 3.1.1. Protocol Review Committee Structure

To relieve any power imbalances between members within protocol review committees (e.g., one community representative versus a majority of university scholars), participants suggested the number of community members be increased, with the intention of increasing the diversity of views. One participant noted that “*...at a committee level, it’s more important to have a diversity of views and actors instead of just having more numbers and expect that greater numbers will bring diversity.*”

Participants also discussed the need for improved recruitment methods to avoid institutional bias in the selection of committee members. As an example, in Sweden local political parties and animal welfare organizations select laypersons to represent community values on ethics review committees. It was also noted that the protocol review process should be independent from the institution where research takes place: “*We need more thought on the committee structure to make [for a] more independent review process.*” In Sweden, the protocol review committee is not affiliated with the institution where the protocol will be implemented, but this was not the case in countries represented by other participants. In all other countries, the protocol review committee is directly affiliated with the institution, and community member selection is less democratic (e.g. new community members selected because they were acquaintances of existing committee members). 

These sentiments echo those highlighted in previous literature (e.g., [29,30]). In particular, Schuppli and Fraser [31] found that, among other factors, the effectiveness of ethics review committees was influenced by committee composition and dynamics, recruitment of members and member turnover. Schuppli and Fraser [31] highlight a potential bias towards institutional or research interests. This bias results from: (1) membership being dominated by scientists, (2) poor leadership by chairpersons that prevented full participation of community and minority members, which leads to poor committee dynamics, (3) “community members” being affiliated with the institution and (4) the motivation of some members to pursue a personal agenda rather than adhere to the committee’s mandate. Other research has looked at the importance of committee structure but considered the role of sub-committees and whether and how these can maximise opportunities for ethical review [32]. 

#### 3.1.2. Protocol Review Committee Function

There was a discussion on the need for a robust and informed decision-making process; specifically, a process that people can trust. It was recognized that some decisions to allow animal research to proceed might not be in line with societal values, so there is a need to engage the local and national community in a broader dialogue related to animal research.

“*…there’s a lot riding on the people who actually sit around the table. Whether they’re the researcher, or the community rep, or the veterinarian, or the Chair, there’s a lot tied to that. Individual history, personal experience, personality, lots of things like that. So how do we get to a place where we have a robust decision-making process that is also kind of, you know, almost repeatable?*”

“*I don’t know if the analogy of the jury system might work…if you took the same case to 10 different juries, you’d get different decisions, right? And somehow, for me anyways, that doesn’t bother me. But the case [of] the animal ethics committee maybe bothers me a little bit more. And I think that’s because I’m somehow less trusting that the process being followed in that community is really, fairly reflects the, well, that the process works …*”

Participants specifically discussed the role of the protocol review committee Chair and highlighted the view that the Chair must provide the leadership to empower all members of the committee to voice their concerns: “*We need the right leadership present on a committee to take charge and to speak up on the ethical issues on the protocols and to keep asking the right questions. So, we need a good Chair to allow consensus regarding the protocols we use for approval and rejection.*” This was considered especially important when trying to ensure that community representatives are fairly included in the decision-making process and are not overwhelmed with jargon or technical details.

It was generally agreed that review of animal research requires a broader value judgment, not just a review of the technical details or the impacts on the animals involved. Broader ethical review may be limited by the expertise of the members or by committees specifically asking members not to review the social value of the research and only focus on harm mitigation. It has been noted elsewhere that the protocol review process typically focuses more on technical details than on a broader value judgment [33]. The call for a broader value judgment has also been made elsewhere [34,35].

It was also generally agreed by the participants that there should be increased openness from protocol review committee members (and institutions) about: (a) the functioning of the committees and how decisions are arrived at and (b) approved protocols. Participants largely agreed that committee members should not be bound by confidentiality and should be able to share information outside the committee (with the exception of proprietary information). This is particularly important for community representatives who may need broader support for their role in the committee. There was also a suggestion to build on the UK model of publishing lay summaries to create a searchable online resource for all approved protocols, including the numbers and types of animals involved. Providing the public an opportunity to engage and comment on proposed animal research utilizing lay summaries has been tested in Canada [36].

Some researchers may have concern about the public release of certain information in their protocol, so there should be some mechanism to redact the protocol with the provision of footnotes justifying the case. It was agreed that certain information should remain confidential, including the names of any research staff. The discussion about keeping names confidential focused on the perceived risk of animal rights activism; this speaks to the tension between openness and perceived security risk that has been discussed elsewhere [2].

In addition, a certain level of confidentiality for ethics review committees might be important so that members feel safe to engage in open conversation with committee members. Overall, there was agreement that some aspects of animal research should be released into the public domain, whereas others could remain confidential. While there was agreement that lay summaries of approved protocols should be made available to the public, no consensus was reached on which specific components should be available. However, it was suggested that openness should be the default, but that confidentiality could be requested in special cases to protect researchers’ personal safety and intellectual property, “*… there should be more of a default to openness, with exceptions for proprietary research that you would write in and say, well, I need to be exempted from this requirement for openness.*”

### 3.2. Institutional Culture

When discussing research institutions, participants tended to talk separately about the need for greater openness within institutions and the need for greater openness between the institutional research community and the public.

#### 3.2.1. Openness within Institutions

Participants discussed how to build a virtuous and self-reflective system at the institutional level to facilitate decision-making throughout the entire process of doing animal-based research. There was general agreement that institutions in all countries need to build an ethical culture where the burden of decision-making is shared between the protocol review committee and the research community (“*we need to share the ethical load…there should be a continuum of ethical responsibilities*”) and that, globally, governance systems for animal-based science require openness, mentorship and a greater sense of connectedness between individuals within an institution. Participants felt that this would be best achieved through a bottom-up approach.

Examples of how to achieve a greater degree of ethical decision-making included providing support (separate from and before protocol review) for young faculty members so that they can initiate their research careers in the field of animal-based research in an ethically reflective manner: “*In a way, what we’re talking about is the…deliberate growing of a network of people who can be helpful in various areas…There’s (potentially) a list of people who volunteer for this and maybe occasionally—I picture them as occasionally—doing an orientation session or maybe helping out [with] a little bit of education for junior researchers and stuff like that.*”

In addition to suggestions for orientation sessions, participants suggested that a core advisory team or supporting office be founded that could guide researchers to other peers who have faced similar problems and to other resources. Participants also identified the need for institutional support in building a network of people who can be called on to help share expertise and ethical practice: “*But there are real, live, people that you’re actually asking to spend their time, and there has to be resourcing for this… the institution that has to support it.*” It is important to note that the suggested network of people is intended to be an “*institutional commitment to changing the culture*”, rather than creating a new bureaucracy within institutions.

It was also emphasized that greater interdisciplinary communication between departments and researchers should be encouraged to support the building of an ethical culture related to animal research. The discussion also highlighted that the “Three Rs” principle (Replacement, Reduction and Refinement, see Russell and Burch [37] for description) does not equal ethical review; rather, the implementation of the Three Rs is part of a larger cycle of ethical reflection. Figure 1 below summarizes this cycle.

Participants generally agreed that increased openness would help to raise standards by allowing greater cooperation between researchers and the institution: “*I do think the evidence is available, but it means you’ve got to have enough openness so that you can try and bring it out and hold it up to scrutiny.*” This encourages the creation of a system that ‘learns’ by having a continuous feedback and evidence-based accountability: “*We have to prove that this system works in a way and to produce the evidence for it.*” Participants recognized the importance of building an ethical culture but also recognized the importance of being clear that this is a learning process and the system will be refined over time. To achieve this goal, cooperation and commitment are needed from all stakeholders within the institution and research community. Elsewhere, it has been noted that openness alone will do little to improve ethical decision-making in animal research [39]. However, as indicated by the workshop participants, an institutional commitment to building an ethical culture of research requires increased openness, not only at the protocol review committee level but at all levels that make up the system of animal research.

#### 3.2.2. Openness between Institutional Research Community and the Public

It was generally agreed that institutions should support the dialogue among stakeholders, including the broader public. There is an opportunity for institutions to issue candid reports that show what research has been approved, the number of animals, degrees of invasiveness, etc: “*…our general idea was that [the evaluation] report doesn’t just get shared with the units in question—in general terms it has to be shared with the larger world…*”

Participants recognized a need for providing information on animal research (in a non-self-serving way) and agreed that there should be opportunities to have public discussions about animal research. Simply publishing information about animal research is not enough, there also needs to be a two-way dialogue where members of the public have the opportunity to have their voices heard:

“*I’m just wondering whether we have a broader understanding of what we mean by openness, what its purpose is…Is it part of the communication, so it’s actually part of a dialogue, or is just a public display about opening doors?...to me it should be in effect a dialogue rather than, you know, we’re having an open house, come around and look and see what we’re doing.*”

One participant highlighted that it is important to keep those involved in animal research out of harms’ way and to protect intellectual property. To this end, it was suggested that public consultation could be carried out to find out what the public would like to know about animal research and the areas in which they would like to see greater openness. This suggestion fits well with the aims of the UK Concordat on Openness on Animal Research [25].

Another way of ensuring greater openness regarding animal research is to invest in meta-analysis so that the benefits achieved by animal research are more accurately known. While methods like systematic reviews were not explicitly discussed by workshop participants, the value of meta-analysis has been documented elsewhere [40]. A more accurate assessment of the harms and benefits achieved by animal research will serve to facilitate proper harm: benefit analysis of proposed research and protocol review strategies that employ a full ethical review rather than just mitigation of harms to animals. This is particularly important in light of a recent study that used a systematic review methodology to retrospectively analyse six approved preclinical animal studies. The study authors, Pound and Nicol, concluded that “All the studies were of poor quality. Having weighed the actual harms to animals against the actual clinical benefits accruing from these studies, and considering the quality of the research and its impact, less than 7% of the studies were permissible according to Bateson’s Cube…” [41].

### 3.3. Policy and Oversight

Participants discussed the ’business of science’ and indicated that funding agencies or granting bodies are an essential part of building an ethical culture of animal research. There was encouragement of further discussions between institutions and granting agencies on developing vehicles for funding agency involvement in the virtuous feedback loop described above.

#### 3.3.1. Openness at All Levels

Participants were in agreement that science in general should be more open, not just animal research: “*the problem isn’t just about the use of animals. It’s also about our openness in science about what we’re producing…this is a smaller part of a much larger picture where we say, oh the system works, but we aren’t willing to hold the system up to public scrutiny and show that it does work, and be self-critical [of] ourselves in the academy.*” However, participants also agreed that when focusing specifically on animal research, creating an ethical culture that is self-reflexive requires openness at all levels (committee, institutions, policy).

“*It’s not just about what we do on animal care committees and the local committees or the national [governing] bodies. It’s a part of a whole systemic thing around public knowledge…put it in the larger context and say we should be trying to justify what we’re doing, and the rest of the system has to do its bit. And if it’s not willing to do its bit, you aren’t going to fix it at the local level, you’re not going to fix it at the national, regulatory level either. There’s got to be something larger going on.*”

Figure 1 illustrates a virtuous, self-reflexive learning loop for animal-based research that goes beyond the accounting of harms to animals that happens in animal ethics committees; it is hard to imagine creating such a system without addressing issues of openness at every stage of the system. 

#### 3.3.2. Findings for Canada

Workshop participants discussed systems internationally, but given the location of the event there was some focus on animal research oversight in Canada. In particular, participants discussed introducing a third level of oversight (as opposed to the current two levels—see CCAC http://www.ccac.ca/en/). Participants describe this potential new system as follows: (1) a national board, responsible for setting minimum acceptable standards and enforcement of these standards by a third party, as well as dedicated animal welfare officers, (2) an advisory animal board to institutions, which involves local or regional stakeholders and the broader community and (3) an institutional board, responsible for internal enforcement, open houses, providing public information on approved protocols, setting internal standards and limits for animal research at that particular institution (which may go above and beyond minimum standards created by the national board). Whether such a system would work will likely depend in large part on the acceptance by those directly involved in animal-based research to incorporating the voices of the broader community; i.e. to what degree transparency and openness are viewed as positive attributes of an ethical oversight system.

In addition to the recommendation for restructuring animal research oversight in the ways outlined above, one participant encouraged the establishment of face-to-face annual meetings at the provincial level. This would allow for greater openness and create the opportunity for dialogue about what animal research practices are deemed acceptable: “*Face to face was very important…There [is] actually time and effort made to invite interested groups to attend, as well as the general public, as well as others, all and sundry.*”

## 4. Conclusions

During the forum, several points of consensus were reached among participants: (a) evidence-based metrics to allow a “virtuous feedback” system for evaluation and quality assurance of animal research, (b) the need for increased public access to information together with opportunities for stakeholder dialogue about animal research, (c) a greater diversity of views to be represented on decision-making committees, to allow for greater balance and (d) a standardized and robust ethical decision-making process that incorporates some sort of societal input.

Arriving at these policy suggestions was not easy; these emerged out of several days of robust discussion. We hope that this paper encourages further discussion on issues of governance and practice that go beyond openness. Academics and stakeholders based in the UK recently undertook a dialogue process in order to identify priorities for further research [42]. The advantage of shared agenda setting processes is precisely that they are co-created to ensure the broad representation of different experiences and perspectives.

The points of consensus from our meeting underscore the productivity of collaborative dialogue, as well as how different perspectives bring strength to the policy suggestions presented. Regulatory frameworks and the level of openness vary between jurisdictions, but these differences create opportunities to share best practices and to move the global discussion for increased public openness forward.

## Figures and Tables

**Figure 1 animals-09-00622-f001:**
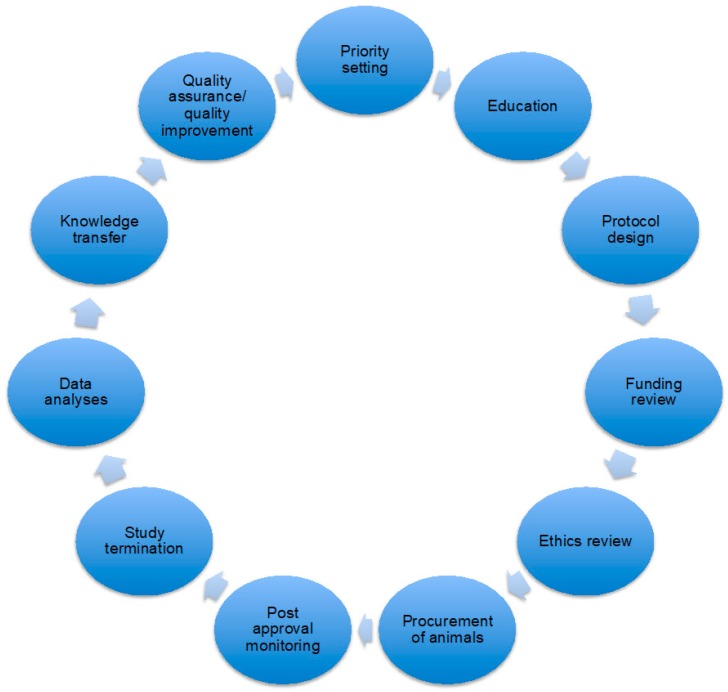
Proposed virtuous, self-reflexive learning loop for animal-based research that would focus on ethical reflection. Ethical reflection would no longer be limited to the ethics review component (adapted from [38]). Animal manipulation occurs from the procurement of animals to the study termination, so it represents a relatively small part of the process of animal-based research.

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
