# Peer review of "Animal Research, Accountability, Openness and Public Engagement: Report from an International Expert Forum"

_animals, 2019, doi:10.3390/ani9090622_

Round 1
Reviewer 1 Report
The research puzzle addressed by the authors of this manuscript is highly relevant for researchers using laboratory animals. The balance between transparency and confidentiality in the use of animals in research is an important question and the question on how to engage the public in the issue remains disputed. Therefore, I believe that the contribution is well suited for the journal animals and should be of interest to the readers of the journal.
The questions developed in the manuscript present certainly an excellent opportunity to develop avenues for future research. The manuscript is not a research article in the original sense, but it does provide interesting starting points for future research. Given the specific character of the manuscript as a commentary I can only make some minor suggestions for improvement:
The arguments that are given for the country selection remain unclear, at least from a political science perspective. The authors refer to different “governance styles”, what exactly do they mean hear? Within the governance literature there are many different strands, so it would be helpful if the authors would specify what they refer to and if they refer to governance styles in the field of animal experimentation or in the political system as a whole.
In the results sections the authors briefly discuss the structure of the review committee (3.1.1) and give only Switzerland and Sweden as examples. I think it would be interesting to get information on how these committees work in the other countries and if there is an independent review process, as the authors suggest. Section (3.1.2) provides an interesting, but rather lose overview of the function of the review committee and the importance of selecting the members. When reading this section, I was wondering if the authors could supply a systematic overview of the status quo in their countries. In section 3.2 the authors reflect on the openness within research institutions and discuss some challenges. This section could likewise be strengthened by providing a more systematic overview of the situation in the countries or within different institutions. I was wondering if variation exists between different institutions within one country or rather between countries and what factors enable or inhibit openness. The same question arises for section 3.2.2 with regard to the openness between institutional research community and the public. Section 3.3.2 then highlights some findings for the case of Canada, here again, I think this section would benefit from a comparative approach to identify similarities and differences between countries.
Author Response
Reviewer 1
The research puzzle addressed by the authors of this manuscript is highly relevant for researchers using laboratory animals. The balance between transparency and confidentiality in the use of animals in research is an important question and the question on how to engage the public in the issue remains disputed. Therefore, I believe that the contribution is well suited for the journal animals and should be of interest to the readers of the journal.
The questions developed in the manuscript present certainly an excellent opportunity to develop avenues for future research. The manuscript is not a research article in the original sense, but it does provide interesting starting points for future research. Given the specific character of the manuscript as a commentary I can only make some minor suggestions for improvement:
AU: Thank you!
The arguments that are given for the country selection remain unclear, at least from a political science perspective. The authors refer to different “governance styles”, what exactly do they mean hear? Within the governance literature there are many different strands, so it would be helpful if the authors would specify what they refer to and if they refer to governance styles in the field of animal experimentation or in the political system as a whole.
AU: We have now adjusted the text to read: “The aim was to recruit participants who represent those countries with a range of governance systems for the use of animals in science, so participants from the US, UK, Canada, Sweden, Australia, New Zealand and Germany were recruited.”(Lines 133-134; highlights indicate where a change was made)
AU: Since the original text goes on to explain what we mean, e.g. participants came from countries with governance systems for animals in science that range from fully legislated, partially legislated to non-legislated, and from governance systems that have varying degrees of openness, we hope that the addition to manuscript Line 134 to clarify that we are not referring to overarching political systems, but only to systems that specifically govern the use of animals in science, is sufficient.
In the results sections the authors briefly discuss the structure of the review committee (3.1.1) and give only Switzerland and Sweden as examples. I think it would be interesting to get information on how these committees work in the other countries and if there is an independent review process, as the authors suggest.
AU: We have now added text to clarify that our focus on Sweden (though not Switzerland, as the reviewer stated) as a case example was because Sweden is unique in having arms-length protocol review committees: “This is not the case in countries represented by other participants. In all other countries the protocol review committee is directly affiliated with the institution, and community member selection is less democratic, with new community members often selected because they are acquaintances with existing committee members. Therefore, Sweden is unique in having arms-length review committees and in using political parties and animal welfare organization to select community members. “(Lines 192-197).
Section (3.1.2) provides an interesting, but rather lose overview of the function of the review committee and the importance of selecting the members. When reading this section, I was wondering if the authors could supply a systematic overview of the status quo in their countries.
AU: To clarify, Section 3.1.2 does not speak directly to the importance of selecting protocol review committee members – that was discussed in section 3.1.1. so we have opted to edit the former section to respond to this comment. See our response above.
In section 3.2 the authors reflect on the openness within research institutions and discuss some challenges. This section could likewise be strengthened by providing a more systematic overview of the situation in the countries or within different institutions. I was wondering if variation exists between different institutions within one country or rather between countries and what factors enable or inhibit openness.
AU: Despite the differences in governance systems for the use of animals in science between the countries represented, the factors that enable or inhibit openness are similar. Therefore, we do not feel there is too much merit in focusing on merely describing the status quo, since one of our key objectives as to outline ways to move beyond our current situations. In preparation for the forum all participants gave a descriptive summary of their governance system for animal-based science, so those summaries were the jumping off point for the discussions presented here. We have hopefully addressed the reviewers concerns and comment there by making the following edits:
Lines 275-280: “There was general agreement that institutions in all countriesneed to build an ethical culture where the burden of decision-making is shared between protocol review committee and the research community(“we need to share the ethical load…there should be a continuum of ethical responsibilities”), and that, globally, governance systems for animal-based science require openness, mentorship, and a greater sense of connectedness between individuals within an institution.”
(highlight shows where changes were made)
The same question arises for section 3.2.2 with regard to the openness between institutional research community and the public. Section 3.3.2 then highlights some findings for the case of Canada, here again, I think this section would benefit from a comparative approach to identify similarities and differences between countries.
While we appreciate the comment, we feel we have already given suitable justification for briefly focusing in on Canada at the end of the paper. This was a Canadian-funded initiative and the forum aimed to bring international people together to brainstorm ideas so that the results could specifically influence Canadian policy. Forum members can also bring these ideas to their respective countries as they see fit. In addition, the participants at the meeting actively discussed the Canadian context, not other countries, and we need to remain faithful to the conversations that happened at the forum.
Reviewer 2 Report
Thanks for the opportunity to review this paper, which is a sort of "focus group" for policy-making.
Although I am not an expert on ethical issues in animal research, I have found the paper instructive and contributing to an important debate, which needs to take into account many aspects, among which the security of researchers as well as accountability to the public funders.
In what follows, I would like to offer a handful of minor comments to the authors:
- on lines 62 and 63: I would not use the verb "allow", considering that I believe no-one wants to let in the position to be the target of potentially violent behaviour. I would re-phrase the sentence
- I appreciate that you have included in the paper the verbatim citations of the speakers but I was wondering whether it would be possible to do minor editing activities to them, such that they could be read more easily
- I believe the reader would prefer to read something about the "Three Rs" referred to on lines 291-292 instead of having to source a very old publication, which may not be really possible after all
- Regarding figure 1, I would make it a bit smaller, to make sure the associated caption stays all close to the Figure and on the same/in one page
- Please check the beginning of line 300 where "Overall" and "generally" are close to each other (maybe too many adverbs?)
- Please check lines 335 to 338 because I am not sure that the phrase delivers the message it was intended to
- The paper focuses quite a bit on "openness", throughout the contribution. However, it seems to underplay the role of "communication", and possibly "communicating in a language that the layperson can understand". Besides the outright "openness" of the procedure, I believe that there is also a significant contribution which could be provided by the "correct communication". There may be scope to discuss this dichotomy in the paper
Author Response
Reviewer 2
Thanks for the opportunity to review this paper, which is a sort of "focus group" for policy-making.
Although I am not an expert on ethical issues in animal research, I have found the paper instructive and contributing to an important debate, which needs to take into account many aspects, among which the security of researchers as well as accountability to the public funders.
AU: Thank you!
In what follows, I would like to offer a handful of minor comments to the authors:
- on lines 62 and 63: I would not use the verb "allow", considering that I believe no-one wants to let in the position to be the target of potentially violent behaviour. I would re-phrase the sentence
AU: The sentence has been rephrased to read: “Historically, sharing details about animal research potentially put institutions and individuals in a more vulnerable position by opening them upto be the target of animal rights activism” (Lines 61-63; highlights indicate where a change has been made)
- I appreciate that you have included in the paper the verbatim citations of the speakers but I was wondering whether it would be possible to do minor editing activities to them, such that they could be read more easily
AU: In qualitative research like this is not common practice to adjust verbatim quotes made by participants because we need to stay faithful to the comments, issues and themes raised by participants – adjusting quotes to make them more readable might inadvertently change the content slightly, so while we respect your comment we have elected to present the people’s verbatim comments.
- I believe the reader would prefer to read something about the "Three Rs" referred to on lines 291-292 instead of having to source a very old publication, which may not be really possible after all
AU: The Three Rs have now been fully listed: “The discussion also highlighted that the “Three Rs” (Replacement, Reduction and Refinement – the principles that guide ethical use of animals in science; see Russell and Burch 1959 for description)does not equal ethical review;…”(Lines 300-301; highlighted text indicates where changes were made)
AU: For those in the field of Three Rs work the 1959 book by Russell and Burch is a seminal text and referred to often. As scholars we feel it important to always go to original source material so we are not going to give refer to an alternate source here. The 1959 text is freely available online, so we have included a link to the full, free online text version.
AU: Citation now reads:
AU: Russell, W.M.S. and Burch, R.L. (1959). The Principles of Humane Experimental Technique, 238pp. London, UK: Methuen. Full text available online at: http://altweb.jhsph.edu/pubs/books/humane_exp/het-toc
- Regarding figure 1, I would make it a bit smaller, to make sure the associated caption stays all close to the Figure and on the same/in one page
AU: Figure has been made smaller – though a note that figure formatting and type setting is typically done by journal editors.
- Please check the beginning of line 300 where "Overall" and "generally" are close to each other (maybe too many adverbs?)
AU: We have deleted the word “Overall” so the sentence now reads: “Participants generally agreedthat increased openness would help to raise standards by allowing greater cooperation between researchers and the institution” (Line 311; highlighted text indicates where a change was made)
- Please check lines 335 to 338 because I am not sure that the phrase delivers the message it was intended to
AU: Sentence has been rewritten to now read: “Another way of ensuring greater openness regarding animal research is to invest in meta-analysis so that the benefits achieved byanimal research are more accurately known.” (Lines 343-344; highlighted text indicates where change was made).
- The paper focuses quite a bit on "openness", throughout the contribution. However, it seems to underplay the role of "communication", and possibly "communicating in a language that the layperson can understand". Besides the outright "openness" of the procedure, I believe that there is also a significant contribution which could be provided by the "correct communication". There may be scope to discuss this dichotomy in the paper
AU: Great comment, thank you. We agree, however, the paper discussion needs to be anchored in what was discussed by participants, and since “correct communication” was not raised in the small or large group discussions there is no strong rationale for including it here.
Reviewer 3 Report
This is a well written communication on an interesting topic. It contributes to the literature and provides some insights on international views on openness in animal research.
My main comment and suggestion is to also provide more information on the participants of this forum. If names of the participants are confidential then please provide a description of the expertise composition of the forum i.e. how many academics (senior posts?); how many in policy-making and what is the nature of their job, etc.
Author Response
Reviewer 3
This is a well written communication on an interesting topic. It contributes to the literature and provides some insights on international views on openness in animal research.
AU: Thank you!
My main comment and suggestion is to also provide more information on the participants of this forum. If names of the participants are confidential then please provide a description of the expertise composition of the forum i.e. how many academics (senior posts?); how many in policy-making and what is the nature of their job, etc.
AU: Each participant is a co-author and their institutional affiliation is listed. We have added a line in the methods to indicate how many participants are academics versus those working in policy:“Of the 11 participants, 7 have primary roles in academia while 4 have work in policy; however, all the academics involved have also contributed to either regional or national policy making processes.” (Lines 130-132)